# Mindset and Reflection—How to Sustainably Improve Intra- and Interpersonal Competences in Medical Education

**DOI:** 10.3390/healthcare11060859

**Published:** 2023-03-14

**Authors:** Lisa Lombardo, Jan Ehlers, Gabriele Lutz

**Affiliations:** 1Didactics and Educational Research in Health Science, Faculty of Health, Witten/Herdecke University, 58455 Witten, Germany; 2Psychosomatic Medicine and Psychotherapy, Faculty of Health, Witten/Herdecke University, Gemeinschaftskrankenhaus Herdecke, 58313 Herdecke, Germany

**Keywords:** intra- and interpersonal competences, medical education, mindset, quality in medical care

## Abstract

Intra- and interpersonal competences (IICs) are essential for medical expertise. However, the effects of current medical curricula seem to not be sustainable enough, even though poorly trained IICs have negative effects on medical practice. A defensive attitude towards openly addressing personal–professional challenges seems to hinder a sustainable implementation of IICs training. Therefore, this study asks about the changeability of IICs and target factors of their implementation in medical education. The aim was to detect factors for the sustainable implementation of IICs in medical education from medical and non-medical perspectives. For this purpose, a total of 21 experts were interviewed. The interview material was analysed according to grounded theory principles to generate core categories to answer the research questions. As a first result, analysis revealed that IICs are changeable and developable, not in all, but in many students. It also showed four central prerequisites for successful implementation: the longitudinal integration of reflection and feedback in medical education and practice; a clear framework and individual path of education; the students’ mindset to develop themselves on a personal level; as well as the superiors’ mindset to openly deal with personal challenges in low hierarchies. Contrasting Carol Dweck’s mindset concept with our findings supports our theory that the development of a mindset which allows an open approach to personal deficits and challenges seems to be of central importance for both students and teachers. Two key factors in this process might be teaching about the impact of mindsets on learning and the willingness of superiors to openly address their personal challenges. To improve IICs in medical professionals, it seems helpful to pay more attention to the development of mindsets. Educating teachers and superiors about targeting factors could be a feasible direction for sustainable implementation.

## 1. Introduction


“*When you are a practising doctor, it is not enough to be satisfied with your theoretical knowledge and neglect the rest. I think as a doctor you have a responsibility not to just say: I am the way I am. But to be open to further self-development.*”(Interviewee-medical student)


### 1.1. Background

Medical guidelines offer scientifically recognised standard approaches for the management of patients. Nevertheless, these approaches need to be adapted to individual needs, as already emphasised by the pioneers of evidence-based medicine [1]. To reasonably weigh individual conditions [2] and improve the physician–patient relationship [3], intra- and interpersonal competences (IICs) are needed to arrive at appropriate solutions [4].

In this context, intrapersonal competences are defined as the knowledge, skills, and attitudes that are important to deal with oneself in a healthy and constructive way, while interpersonal competences are needed to work productively with others [4]. Intrapersonal competences include, for example, self-awareness, reflectiveness, self-regulation, self-care, moral resilience, and the ability to develop ideals and values that foster the development of professional attitudes. Interpersonal competences comprise characteristics such as patient-centeredness, empathy, and constructive relationships with patients, colleagues, and institutions, as well as dealing with mistakes.

The described definition of IICs is used as a collective term for various competences defined by the National Academy of Science, Engineering, and Medicine [4]. Because research and teaching more often focus on individual competences than on their commonalities as IICs, there has been little research devoted to this “collective term.” Nevertheless, in practice IICS are needed in varying combinations to improve the quality [5,6,7] and outcome of medical treatments [8]. Furthermore, well-trained IICs increase patient safety [5,9,10,11,12,13] and satisfaction, as well as healthcare providers’ satisfaction and health [14]. 

Even though there is sufficient knowledge, well-structured skills training, and frameworks for developing personal competences [15,16,17], IICs still need improvement in many places in medicine [18,19]. However, instead of improving over the course of medical training, IICs tend to decrease—especially during the clinical training phase [20,21]. One reason for this could be that IICs cannot be developed sustainably through brief interventions, theoretical courses, or skills training alone [15,18,22,23]. However, there are methods and interventions known to more sustainably implement IICs: in addition to knowledge and skills training, the longitudinal reflection and feedback on a personal level, e.g., on one’s motives or values, are known to develop, refine, and sustain IICs [24], especially in clinical settings.

However, despite this evidence there seems to be a reluctance to implement communicative and reflective interventions to improve IICs in the clinical setting, fostered by a defensive mindset towards dealing openly with mistakes and shortcomings at a personal level [19]. Mindset is defined as an “established set of attitudes”, as “the outlook, philosophy, or values of a person”, and “frame of mind, attitude, [and] disposition.” [25]. Within her broad research, Carol Dweck defines two types of mindsets: the fixed mindset and the growth mindset [26]. The defensive mindset on a personal level described above can be linked to the concept of a fixed mindset [26]: an attitude that views intelligence and competences as something static and therefore sees personal challenges and mistakes as threats rather than learning opportunities [26]. In the previous part of this study, we could find out that in medicine, this fixed mindset becomes a barrier for the implementation of IICs as it seems to be not only a personal attitude, but a cultural trait influencing different aspects of the medical professions [19]. This cultural barrier has an origin in healthcare teams having a “belief that quality of care and error free clinical performance are the result of being well trained and trying hard” [6]. From this perspective, errors become personal failures and addressing them openly becomes a challenge [6]. This belief becomes a cultural barrier as “organisational culture represents the shared ways of thinking, feelings and behaving in health care organisations [7]”. Therefore, a closer look at the central aspect of the culture of a health facility and addressing medical teams and organisations as a whole rather than individuals only is essential for improving the implementation of IICs and thus the quality of care [6,19].

Failure to achieve sustainable implementation and low IICs can have negative consequences, not only in direct patient care, but also in various areas of medical practice, such as teamwork, leadership, and personal professional development, as well as in the quality and safety of medical care [5,11,12,13,18]. However, there are examples of implementation strategies for IICs development in teaching, training, and work routines in medical and non-medical professions: In aviation, for example, a high percentage of accidents used to be caused by human error [27] due to poor communication [28]. Targeted training in error communication within crews and the promotion of feedback structures in all hierarchical directions reduced avoidable errors [11,27,28,29]. Feedback is reported to “increase effort, motivation or engagement […]” [30]. In addition, reflection is often cited as an important tool to enable learning from personal experiences [31]. Research on teacher education shows that a teacher’s experience does not seem to be as important as his or her ability to reflect on his or her interaction with students and to apply this knowledge flexibly [31].

To achieve culture change within medical teams, the promotion of interpersonal skills was integrated as a “third post of specialist training” [32], intending to create a culture that allows for an open approach to mistakes, challenges, and other issues [33]. Trainings to improve “communication, leadership, decision-making, conflict resolution, stress and stress management” in the clinical setting were implemented [34]. Based on these and other positive examples, there is evidence of positive outcomes from this implementation in terms of increased safety as well as reduced costs in medical practice [27,28,35].

Knowing about these positive examples, indications, and developments raises the question of why a cultural change regarding the implementation of IICs in medical education and practice is still difficult to achieve in many places. This leads back to the general question of whether IICs are modifiable or whether they represent a static personality trait. In personality research, it is assumed that personality traits are modifiable throughout the lifespan, influenced by life experiences and goals as well as by the way of life someone has chosen [36,37]. Carol Dweck has also shown in her extensive research that the fixed mindset can be transformed into a growth mindset. While the fixed mindset prevents development, students with a growth mindset see challenges and mistakes as learning opportunities [26]. She was able to show that—in general and with interindividual differences—this type of mindset leads to improved learning in schools and colleges and better learning outcomes [4,26,38,39].

### 1.2. Research Questions

Efforts to date are not yet sufficient to sustainably teach the necessary IICs for high-quality, patient-centred care, even though IICs are supposed to be changeable, and interventions exist which can change them. At this point, it should be said that this publication is the second part of a qualitative study. The first part was a barrier analysis and was published in 2019 [19]. In this first part, we identified factors that are hindering the implementation of IICs. These factors are best subsumed as a fixed mindset culture.

The relevance of our work against this arises from the assumption that decades of research on the role of IICs as well as the addition of training of individual IICs to medical curricula have so far not led to a sustainable implementation of IICs in medical teaching and practice. After identifying cultural obstacles [19], we would now like to present whether and how these could be overcome.

Thus, the questions remain: do experts consider IICs in the medical context to be changeable and what levers can be used for implementation strategies in the face of barrier analysis?

Our aim was neither to focus on teaching individual competences nor to develop actual recommendations for specific action in practice. We wanted to stay at a general level, to look at IICs as collective term, and focus on strategies to achieve cultural change. Against this backdrop, we asked medical and non-medical experts the following:Can intrapersonal and interpersonal competences be developed in medical education or are they a static personality trait?What are the most important factors in implementing the development of intrapersonal and interpersonal competences in medical education?

## 2. Materials and Methods

### 2.1. Study Design

Since intra- and interpersonal competences (IICs) and their training depend on inner processes and the handling of motives, expectations, mindsets, feelings, and attitudes, the diversity of experiences, inner beliefs, and assumptions had to be considered in the design and methods of the study. Therefore, a qualitative research method with semi-structured interviews was chosen. To create transparency in the qualitative research processes, a completed COREQ checklist can be found in the Appendix A.

On 29 August 2016, the ethics committee of Witten/Herdecke University adjudicated that there were “no ethical or professional concerns” regarding the implementation of this study (No. 120/2016).

### 2.2. Sampling

To map the complexity of strategies and barriers in implementing IICs from a wide range of perspectives, a heterogeneous group of interviewees was formed by means of theoretical sampling [40]. The aim was to capture common ways of thinking and deeper shared—perhaps even partly pre-conscious—assumptions, while exploring the role of IICs within the medical (education) culture and in particular capturing ways to overcome barriers to the implementation of IICs education in the clinical setting. As these characteristics are sometimes difficult to capture for people who are themselves part of the medical community, we also included interviewees with a non-medical professional socialisation.

When selecting the interview partners from the medical field, the aim was to create a meta-perspective on medical culture in general. Therefore, we included various stakeholders and professionals from both surgical and conservative disciplines, as well as experts from research, medical education, and practice. Another idea to broaden the group of interviewees was to map supposedly different mentalities regarding the research topic within the medical culture: on the one hand, specialties such as psychiatry and psychotherapy, which are known for their focus on reflection, supervision, and thus on the training of IICs, and on the other hand, specialties that are assumed not to be defined by this focus, such as surgery, orthopaedics, or emergency medicine. As the training of IICs ideally involves the entire medical team, the study also included nurses and medical students.

When selecting non-medical experts, we placed special emphasis on professional fields in which a high level of IICs is required and which offer in-depth IICs training, e.g., social workers, teachers, but also professionals from aviation and management. These experts from the non-medical fields had in common that they also had professional experience in the medical field. In addition to these professional backgrounds, experiences from their own patient situation or from supporting relatives during treatment in different medical facilities were also included. For this reason, no interviews were conducted with patients only.

Another selection criterion was the inclusion of international perspectives on the implementation of IICs. While most interviewees were from Germany but worked in international contexts, there were also participants from Belgium, Austria, Israel, and the USA. We included men and women with different levels of training, ranging from medical students to highly experienced practitioners.

Using the sampling criteria described above, the selection of interviewees was repeatedly discussed and expanded until the interview material reached a saturation of content and no new perspectives could be found in the interviews.

All interview participants involved in the study took part voluntarily and agreed to the publication of the data in an anonymous form. The welfare and protection of the participants were guaranteed by respecting their rights, privacy, dignity, and sensitivities. The Helsinki declaration was adhered to, including, but not limited to, a guarantee of participant anonymity and obtaining the written informed consent of participants.

### 2.3. The Interviews

In the first phase of the research project, an interview guide was developed according to our research questions. The interview guide was developed based on the research questions and against the background of the current state of research. It was tested by conducting two think-aloud interviews and then revised to refine the interview questions. The English version, translated from the original German, was edited by a native English translator.

At the beginning of each interview, the definition of IICs was given (Appendix A questionnaire). The definition given in the interview guide does not claim to be exhaustive but served to find a common basis for the interview questions and to provide comprehensible examples of IICs. The complete interview guide can be found in the Appendix A.

The interviews were conducted in a semi-structured process using the interview guide in English or German in person or by telephone, depending on the interviewees’ native language. A total of twenty interviews were conducted with twenty-one interviewees between June 2016 and March 2017.

Ten men and eleven women were interviewed for a total of 15.9 h (between 16 and 85 min per interview). The interviewees were aged between 23 and 70 years and their work experience ranged from 0 to 46 years. They came from the USA (1), Belgium (2), Israel (1), Austria (1), and Germany (16). Please refer to the Appendix A for the details of the interviewees’ demographics. 

Each recorded interview was then transcribed and anonymised. The quotes from the German interviewees used in the text were translated by a native English speaker.

### 2.4. Analysis of the Data

At the time of the interviews, JE had a DVM and PhD, was a professor, and directed an institute for medical education research. GL had an MD and PhD, she was a psychosomatic medicine physician, taught reflection and personal professional development, and was active in research in this area. LL was a PhD student and involved in organizing a mentoring program. All three researchers read the transcribed interviews separately. The material was then analysed using grounded theory principles [40,41] assisted by the qualitative data analysis tool MAXQDA [42].

In the first step, open coding, the interview material was analysed with the aim to find repeating themes by thoroughly reviewing the data and coding emergent themes with keywords. In the next step, axial coding, the codes were grouped into concepts and then categorized by looking for relationships between the emerging themes. Finally, in the third step, the categories created through this process, as well as the links found between them, were used as the basis for the development of new theory through selective coding. Please refer to the Appendix A for the details concerning the steps of coding. 

During the analysis process, all three researchers discussed the material and the codes regularly and iteratively until a consensus was reached [40]. To move from the initial raw material to theory the researchers moved back and forth through constant comparative data analysis. Two researchers developed the open coding while the third researcher held a more external perspective: his task was to look for relationships, contradictions, and emerging themes from the outside perspective, especially during axial and selective coding.

### 2.5. Sensitizing Theories 

After the main categories had emerged in the described process of analysing the interview data, the researchers looked for sensitising concepts to contrast these main categories and to deductively test the inductively obtained results. One main concept used in this contrasting process was Carol Dweck’s mindset concept. This concept proved to be suitable for repeatedly reviewing, deductively elaborating, and contrasting the inductively obtained results.

### 2.6. Publication Process 

During the analysis process, it became clear that the amount of qualitative data was too large to be covered in one publication, so it was split into two articles. Building on the first part of the research project published in 2019 [19], which dealt with the analysis of barriers preventing the implementation of IICs in medical education and practice, this second article elaborates strategies to overcome the barriers mentioned by the interview participants.

## 3. Results 

The results of the analysed interview material can be divided into two main perspectives: the changeability of IICs and the target factors that are needed to be able to change IICs sustainably. These two main themes are in turn subdivided into sub-categories that identify concrete starting points.

### 3.1. IICs Are Basically Changeable and Developable 

As the first emerging theme, most respondents described intra- and interpersonal competences (IICs) as changeable and developable. They described three different types of medical students: The first type was defined as students who already have a high level of IICs at the beginning of their medical training. In contrast, the second type was defined as those students who will not develop sufficient IICs even during training. The last type was students who can develop IICs throughout medical training if they are well trained. This type of student was considered the most common and the main target for the development of a structured and longitudinal IIC curriculum. Students who are not able to develop IICs during medical school should be identified during medical school selection processes. Only three respondents stated that IICs are something given and stable.
“*A third of physicians just have a natural talent to create this interpersonal aspect well. A third are capable of changing their behaviour when they get the necessary guidance, […] you can still influence them, get them on-board with training and the right ideas. […] one group can’t be reached by training […]. One has to filter them out in advance […].*”(Aviation trainer and aviation safety researcher)

### 3.2. Prerequisites for Successful Implementation 

In the second part of the research results, the individual themes can be summarised under the general perspective of the prerequisites for successful implementation. In this context, the interviewees mentioned several factors that could lead to a cultural change in the implementation of IICs in medical education. These factors relate to the influence on the development of IICs of medical institutions, medical students, supervisors, and medical teachers and are described in detail in the following.

#### 3.2.1. Continuous and Longitudinal Implementation 

Our interviewees reported that IICs are currently not given much attention in clinical medical education, but are rather developed in an unstructured way, depending on the value that medical teachers place on this part of medical expertise. IICs education should therefore be implemented not only in medical schools, but also in medical institutions, so that it is not just an externalised part of education within the broad spectrum of medical degree programmes, but a truly established culture in everyday medical practice. This should be done by implementing IICs in the sense of longitudinal learning for students in medical education and in the sense of lifelong learning for professionals in medical practice. This would allow individuals to learn and grow throughout their professional lives and give the necessary continuity and meaning to the development of IICs.
“*The pre-clinical medical interview curriculum needs to become a curriculum about patient-centred communication, […] the importance of emotions and empathy. In the clinical years, students […] need to be shown the same skills by their supervisors, by the people they look up to when they come to the hospital and the clinic. […] So it’s an ongoing and big task. […] You know it needs to be ongoing training.*”(Medical teacher)

#### 3.2.2. Feedback and Reflection on Personal Development as Key Elements

In addition to the general methods already used in medical curricula, such as knowledge transfer and skills training, as well as the importance of practical experience with real patients and colleagues, our interviewees mentioned reflection and feedback as the key aspects of longitudinal IICs development, as these are indicated as essential for growing on a personal level in a team environment.

Reflection can take place in different settings but should be linked to experiences in medical education and practice. The respondents described different constellations of reflection settings with peers and supervisors, in individual or group settings or portfolio tasks.
“*It is essential […] to provide regular time specifically for reflection in the course of the training. That can be group reflection, that can be student balint groups, that can also be something like reflective writing about certain situations, where space is simply created to promote this aspect of reflection and […] it would be good if this aspect of reflection was built more into the curriculum.*”(Physician and medical teacher)

Feedback should relate to the reality of medical teamwork and reflect the need for different perspectives in medical practice. Feedback should therefore include the perspectives of all team members in all hierarchical directions as well as the perspectives of patients. Feedback needs to be integrated into medical practice in a constructive way so that it is helpful and not seen as another stressor. Those who give feedback should be guided and trained to give feedback in an appreciative and constructive manner.
“*Of course, it would be nice to have structures that really support you in this, […] so that you get regular feedback on where you need to improve, so that you don’t have to pick it up from your own observation.*”(Medical student)
“*I think that it should become a routine to share about what you are doing.*”(Teacher and teachers’ trainer)

#### 3.2.3. Clear Framework and Individual Path 

In terms of promoting the longitudinal implementation of IICs, respondents said that schools and institutions need to clearly communicate their expectations regarding the development of IICs and provide a framework with sufficient space and time, as well as appropriate teaching methods and suitable assessment tools, in medical curricula. Educational institutions need to make firm agreements with their students within this clear framework. However, as the development of IICs is an intimate process in which one is confronted with one’s own emotions, values, fears, and strengths, students should have some freedom to choose their personal curriculum within a given framework.
“*With all these things (the training framework), I always vacillate between the given framework and the space for creativity. I try to do things in a way that an agreement can be reached, but the way an agreement is reached always has to remain a bit individual.*”(Teacher and teachers’ trainer)

This clear framework could lead to the development of IICs being understood as a natural and self-evident part of medical education, just like other knowledge- or competence-based subjects. This equal integration would be an expression of the value of IICs for fruitful medical education and practice. It requires a personal and institutional awareness of the importance of IICs for the adaptation of general guidelines and medical knowledge to the individual person, both in education and in patient care.
“*But I think it would be helpful if (reflection) […] is not denigrated in training as social chatter or something, but that it becomes or is an essential part of medical practice.*”(Lawyer and management consultant)

#### 3.2.4. Students’ Open Attitude towards Personal Development 

In terms of student prerequisites, most respondents described a certain mindset that is necessary to benefit from education in IICs. The key element of this mindset was described as a belief in the need and willingness to develop on a personal level. Medical students therefore require an open attitude defined by motivation, curiosity, and a belief in the need to develop one’s IICs. The interviewees stated that this mindset can be identified during the selection processes for medical schools but should also be trained in medical education.
“*I really believe that anyone who wants to (develop IICs) can. Of course, there are people who may have a different aptitude, a different sensitivity, but still I think that if you pay attention to these subtle levels of conversation and interaction, if you value them, then you are on your way and you can learn a lot. But that takes a willingness and a conviction that it’s important.*”(Consultant neurologist)

#### 3.2.5. Superiors’ Willingness to Openly Deal with Personal Challenges in Low Hierarchies 

In addition to all the prerequisites mentioned above, respondents indicated that for sustainable training in IICs, one of the most important factors to enable culture change is that supervisors themselves become open to feedback and deal openly and constructively with their own mistakes. This open mindset must go hand in hand with flat hierarchies that allow reflection and feedback between different levels of the hierarchy.
“*You know what my trick was? We used to say: Describe your mistakes and we’ll talk about it. That didn’t work. So, in the trainings, instead of saying: describe your mistakes, I started talking about the mistake I made on my last flight. […] And suddenly someone said: Oh, yes, something similar happened to me once. And that’s how you create an open atmosphere. But the most important thing is that the trainer has to lead by example.*”(Aviation trainer and aviation safety researcher)
“*But the most important thing is to train the supervising physicians, the residents and the attending physicians. We need to train the people at the top because the students need to see possible role models and also they need to get meaningful feedback from the senior physicians.*”(Physician and medical teacher)

## 4. Discussion

Examples of successful implementation of intra- and interpersonal competences (IICs) in medical and non-medical professions, as well as evidence of their positive impact, lead to the question of whether IICs can be further developed in medical education and clinical practice, and what factors might promote sustainable implementation in medical curricula and routine clinical practice. To answer these research questions, we interviewed a diverse sample of medical and non-medical experts in education and training in different professional fields.

### 4.1. Can Intra- and Interpersonal Competences Be Developed? 

Respondents indicated that most students could improve during medical training in IICs. Those who cannot improve much through training should be identified in the selection process.

Research also shows that character traits—including IICs—are usually seen as changeable aspects of personality that can develop over a career and life, depending on the type of work and life someone has chosen and the events he or she faces [36,37,43]. Therefore, student selection procedures in aviation [28] and in teacher training programmes [44] use measures to check for personal fit. There are also examples in medicine of the increasing importance of personal characteristics in student selection [8,45,46], as some of these characteristics are seen to “predict quality of patient care” [47].

### 4.2. What Are the Most Important Factors for the Implementation of Intra- and Interpersonal Competences?

The main finding was that the implementation of IICs requires a culture change in the sense that it is valued by institutions and individuals as critically important for the success of medical diagnosis and treatment and for a fruitful physician–patient relationship. To implement the development of IICs as a natural part of medical education, various factors need to be promoted among institutions, students, and supervisors. The development of IICs should be organised in a longitudinal, continuous setting, and feedback and reflection should be integrated as key methods for personal growth. Institutions should be aware of the importance of IICs for educational and treatment success and provide a clear framework for an individual development pathway. Student selection and training should be based on a specific mindset that enables growth at a personal level. As a key component of flat hierarchies, supervisors and teachers should demonstrate that they are open to reflection and feedback and disclose their own mistakes and failures.

As also described by our interviewees, longitudinally integrated feedback and reflection are indicated as crucial for the development of IICs and professional identity [48,49]: they are described, e.g., as “one of the most powerful influences on learning [30]” and part of the most important aspects of professional development [50,51]. However, reflection and feedback for students need to be linked to practical experiences and meaningful personal challenges as development opportunities [26,30,52]. To learn from experiences and challenges, students need an awareness of and reflection on their own [53] and others’ feelings and inner processes [54,55]. For teachers, reflection becomes central when it comes to developing their own role as a teacher, especially regarding the “humanistic aspects of medicine” [56]. The implementation of reflection and feedback can therefore not be limited to the medical curricula of medical schools but must be implemented longitudinally in all areas of medical education and practice, including the clinical workplace, and at all levels of the hierarchy.

With regard to institutions, early and clear communication of the requirements of a profession to students and professionals, as well as a clear framework, are mentioned as fundamental factors for the sustainable development of IICs [4,28]. Students who show difficulties in developing their professionalism need the support of the university to improve. On the one hand, this requires the promotion of IICs. At the same time, however, this also includes clearly defined goals that need to be achieved and named consequences if this does not happen [47]. It is suggested that the organisational goal is changed accordingly in all institutions involved in medical education and structures are developed that provide time, space, and support for development, as well as appropriate assessment tools [22,47,57,58,59].

Many of these factors are already known and implemented in various professional fields. The mindset of the students described by our interview participants can be linked to Carol Dweck’s growth mindset concept, which allows an open approach to deficits and mistakes, as these are used as learning opportunities [26]. They are—like organisational changes and personal challenges—also essential in her concept for changing the prevailing fixed mindset into a growth mindset [26,52] to promote learning and enable the sustainable development of personal and professional competences [4,19].

In support of this view of the role of teachers, our interviewees identified supervisors as crucial to change learning and teaching cultures. Research in other professional fields also discusses the role of supervisors and hierarchy: In aviation, supervisors and instructors were described as particularly prone to making mistakes within structured work processes, as they were the ones who developed these structures and are therefore less likely to be criticised for mistakes [27]. Our aviation interviewee described that asking his students to report their mistakes and teaching them the implications of a lack of openness in dealing with personal shortcomings and mistakes did not work. However, when he as a supervisor started talking about his own mistakes on his last flight, the students became open to reflection. On the one hand, he thus fulfilled the mandate of the teaching professional to constantly develop and educate himself as a teaching person [60]. On the other hand, this example makes it clear how superiors can create a culture of openness by leading the way, which allows staff and trainees to also deal openly with challenges, questions, and mistakes [7]. Flat hierarchies have therefore been introduced in aviation, allowing feedback between all levels of the hierarchy in all directions [33]. Flight crews are trained as teams rather than individuals to be able to recognise personal boundaries, to have standardised communication, and to promote appreciative interaction within the team [33].

There are example projects that have tried to transfer team training methods from aviation to medical teams as they show great overlap in the need to train non-technical aspects [61]. One study showed that team training reduced the “number of preventable adverse events” by about 25% [35,62]. Likewise, absenteeism and terminations were significantly reduced, thus lowering costs. These trainings based on the aviation model are now firmly implemented.

Furthermore, Carol Dweck tries to define a new role for teachers within educational institutions: Teachers should be collaborators for improvement rather than judges of intelligence [52]. They should praise and encourage their students, focusing on processes rather than results [52]. This specific role can be linked to the finding that physicians are seen as excellent role models when they spend a large part of their time teaching and when “the psychosocial aspects of medicine” are included in that teaching [63] and teachers themselves show a growth mindset [64,65]. As our interviewees also noted, clinical teachers identify role modelling as “the primary method […] to teach […] humanistic aspects of medical care” [56]. However, positive role models are named to promote IICs, e.g., empathy [23,66], and their impact does not always lead to sustainable change in the approach to IICs in medicine: the impact of positive role models is limited if they are only applied in short courses [23] or if students encounter negative role models in everyday clinical life [15,51]. Cultural change and the implementation of IICs development must therefore involve teachers and supervisors as role models to promote cultural change.

For another step towards culture change, Carol Dweck mentions a meta-level aspect that was not described by our interviewees but should be part of further discussions and research approaches, namely that teaching about the impact of mindsets is crucial for the development of a growth mindset culture. In their teaching concept, the impact of different mindsets on the enjoyment and success of learning is mentioned as a fundamental step in changing students’ mindsets and learning cultures [52]. Students should be taught through appropriate training that they can change their mindset. Dweck showed that this meta-level teaching about the influence of *growth* and *fixed mindsets* on personal development improved learning outcomes [26,52,67].

## 5. Advantages and Disadvantages of the Study

To our knowledge, our two-part study is the first not to focus on individual elements of IICs (such as communication, empathy, etc.), but to address the question of how to overcome the obstacles underlying the challenge of implementing IICs in medicine in general. It may therefore seem generalised and not always sufficiently differentiated. However, as there have already been many studies on individual IICs elements, a study looking at IICs at a meta-level seemed necessary. This study was also the first to include different non-medical fields and different nationalities to broaden the perspective of the research questions. However, only experts from Western countries were interviewed, so a perspective from other cultural backgrounds is missing. In order to be able to interview experts whom we could not meet for a face-to-face interview, individual interviews were conducted on the phone. We did not feel that this influenced the data collection process but left a note to that effect in the interviews concerned. Another shortcoming could be that there was no interview participant who exclusively brought in the patient perspective. At the beginning of each interview, the interview participants agreed to contribute his or her perspective as a patient or relative. Six of our interview participants explicitly described their own experiences as a patient or patient relative. In addition, one of the interviewed experts is active in political institutions to professionally convey the perspective of patients and patient representatives. While the results were formulated with consistent views from all interview participants, the selected quotes unfortunately do not allow each expert to speak directly. The quotes that best describe the respective category were selected. Thus, individual interview participants were quoted several times, while others do not appear explicitly.

## 6. Conclusions and Outlook

This work addresses the changeability of IICs and approaches of how to sustainably implement IICs training in medical education and practice. The goal was to describe ways to overcome existing cultural barriers to implementing IICs in medicine. Our research was able to confirm the assumption that IICs are fundamentally modifiable and thus can be addressed in the context of medical education and practice.

To achieve this implementation, a longitudinal approach for IICs training is needed. This approach should provide a clear framework and allow individual freedom in the training process. Within this framework, feedback and reflection are key elements for the sustainable development of IICs.

Furthermore, an open attitude of both students as well as supervisors and teachers is essential for a successful IICs training. The openness of teachers and supervisors in dealing with challenges, uncertainty, and mistakes seems to be a key factor in this context. As role models, they can promote a growth mindset culture. Educating them about the detected targeting factors of IICs implementation could be a feasible way to achieve this implementation in a sustainable way.

Changing these factors on an institutional but also personal level could improve IICs in medical care and thus could make high-quality person-centred medicine possible.

Based on these findings, the next important and necessary step should now be taken: the development of strategies, concepts, and curricula for medical education, collaboration, and practice with the goal of establishing a growth mindset culture in medicine and implementing IICs in a sustainable way.

## Data Availability

The datasets used and analyzed during this study are fully available from the corresponding author at any time. Though, the authors decided not to include the whole data within this paper since most data material was collected in German language.

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
