# Peer review of "Mindset and Reflection—How to Sustainably Improve Intra- and Interpersonal Competences in Medical Education"

_healthcare, 2023, doi:10.3390/healthcare11060859_

Round 1

Reviewer 1 Report

Mindset and reflection – How to sustainably improve intra- and interpersonal competences in medical education 

The theme is very interesting since, since the patients are sick people and therefore in a state of vulnerability, a culture of good skills in the institution is essential. The treatment of the doctor is very relevant to the patients. And the way a doctor relates to his patients and colleagues depends on his intra and interpersonal competencies (IICs). The definition of mindset varies between the meanings of “mentality”, “beliefs”, “attitude”; there are more inclusive definitions “a complex mental state involving beliefs and feelings and values and dispositions to act in certain ways”. Mindsets is a construct that arose along the lines of the findings of Robert Sternberg and Howard Gardner. The authors of this article draw on Carol Dweck's “fixed mindset” and “growth mindset” concepts and how mindset research examines the power of such beliefs to influence behavior and affect challenge seeking and resilience. 

The main perspectives that we value in the authors is the consideration of cultural change in the implementation of IICs in medical education, intending

to create a culture that allows for an open approach to mistakes, challenges, and other issues. Continuous implementation could establish culture in everyday medical practice, using feedback and reflection on personal development as key elements.

The main finding expresses that the implementation of IICs requires a culture change in institutions and that the process needs to be linked to practical experiences and meaningful personal challenges as learning opportunities. The IICs must be integrated into the medical curricula of medical schools and implemented in all areas of medical education and practice, including the clinical workplace, and at all levels of the hierarchy if an authentic culture of care is sought. We agree with the authors on the need for organizational changes and personal challenges to promote learning and enable the sustainable development of personal and professional competencies.

This cultural change requires the design of new curricula and appropriate working environments. In this sense, the authors have not indicated proposals. Nor was the need for collaborative work among colleagues mentioned in the interviews to achieve this change in professional culture.

The references are well focused but could be updated better. The methodology is developed. And in general, it is a coherent study in its sections. Even though it is a qualitative research, a more dynamic and structured presentation could have been achieved, less repetitive and with specific implications on the design of appropriate practices to achieve cultural change.

Reviewer 2 Report

I understand that due to the size of the text, the model questionnaire was not included. This is a pity, because a description of the research procedure would have been more full. But this does not affect the substantive quality of the article.

Reviewer 3 Report

A section on theoretical content should be included after the introduction  in order to contextualize the study. It should  include other  similar studies and mention why the research is relevant.

Section 2  (Materials and methods) should  also reorganized as  it is confusing as it  is. Maybe  it should include the information on participants, data and method in a more structured way.

Reviewer 4 Report

An appropriate work for an important topic. A bibliography adecuate and good use of intruments. Methodology should be defined with more details
